# Investigating the Impact of NMDA Receptor Organization and Biological Sex in the APPswe/PS1dE9 Mouse Model of Alzheimer’s Disease

**DOI:** 10.3390/ijms26041737

**Published:** 2025-02-18

**Authors:** Senka Hadzibegovic, Bruno Bontempi, Olivier Nicole

**Affiliations:** 1Neurocentre Magendie, INSERM U1215, 33077 Bordeaux, France; senka.hadzibegovic@gmail.com; 2University of Bordeaux, 33077 Bordeaux, France; bruno.bontempi@u-bordeaux.fr; 3Institut de Neurosciences Cognitives et Intégratives d’Aquitaine, CNRS UMR 5287, 33000 Bordeaux, France; 4Institut Interdisciplinaire de Neurosciences, CNRS, UMR 5297, 33077 Bordeaux, France

**Keywords:** NMDA receptors, extrasynaptic, PSD-95, memory, synaptotoxicity

## Abstract

Alzheimer’s disease (AD) is a neurodegenerative disorder characterized by memory loss and cognitive decline, with women being disproportionately affected in both prevalence and severity. A key feature of AD is synaptic loss, particularly around amyloid-β (Aβ) aggregates, which correlates strongly with the severity of dementia. Oligomeric Aβ is believed to be the primary driver of synaptic dysfunction by impairing excitatory neurotransmission through interactions with synaptic receptors, including N-methyl-D-aspartate (NMDA) receptors. However, the influence of sex on these synaptic changes and NMDA receptor mislocalization in AD is not well understood. This study examined potential sex-specific differences in synaptotoxicity and the role of extrasynaptic GluN2B-containing NMDA receptors in AD pathogenesis using the APP/PS1 double transgenic mouse model. Although both male and female mice showed a similar amyloid burden and cognitive impairments, synaptic alterations were slightly less severe in females, suggesting subtle sex differences in synaptic pathology. Both sexes exhibited the mislocalization of GluN2B subunits to extrasynaptic areas, which was linked to reduced PSD-95 levels and the synaptic accumulation of Aβ_1–42_. Intrahippocampal injections of DL-TBOA confirmed the role of extrasynaptic GluN2B-containing NMDA receptors in memory dysfunction. These findings emphasize the importance of targeting synaptic receptor trafficking to address AD-related memory deficits, potentially offering a therapeutic approach for both sexes.

## 1. Introduction

Alzheimer’s disease (AD) is a progressive neurodegenerative disorder marked by cognitive decline and memory loss. It exhibits a notable sex-specific profile, disproportionately affecting women in both prevalence and severity [1,2,3]. Synaptic loss, an early and consistent hallmark of AD, is strongly correlated with the severity of dementia in patients [4,5]. This deterioration is particularly pronounced near aggregated amyloid-β (Aβ) or amyloid plaques [6], implicating Aβ in this selective process [7]. Although the exact mechanisms by which Aβ drives synaptic dysfunction and loss remain unclear, growing evidence points to the oligomeric form of Aβ as the primary driver. These oligomers interact with the postsynaptic density, either directly or through synaptic receptors, triggering signaling cascades that impair synaptic plasticity and lead to synaptotoxicity by disrupting excitatory neurotransmissions [8,9]. Epidemiological studies further reveal a higher prevalence of AD among women even when controlling for age, suggesting that additional biological factors such as sexual dimorphism in brain structure, hormonal signaling, genetic risk factors, or immune responses likely play a role in this gender disparity [3,10]. Despite these findings, the influence of sex-dependent alterations on AD remains poorly understood.

Since the discovery that N-methyl-D-aspartate (NMDA) receptors play crucial roles in both cellular models of learning and synaptotoxicity, their dysfunction has been recognized as a contributing factor in the pathophysiology of AD [11]. Aβ disrupts NMDA receptor function through several mechanisms. First, Aβ can interact directly with NMDA receptors or indirectly through intermediaries targeted by Aβ [12]. Second, NMDA receptors may play a pivotal role in mediating or facilitating the effects of Aβ on synaptic transmission and plasticity [13,14,15]. Third, NMDA receptors are significant downstream targets of Aβ, which reduces both their surface expression and function, particularly affecting GluN2B-containing NMDA receptors [16,17,18,19,20,21,22,23]. Fourth, Aβ diminishes the levels of PSD-95 [24,25,26,27,28,29], a key postsynaptic scaffold protein, thereby impairing the synaptic anchoring of NMDA receptors [30]. This mislocalization can result in abnormal tonic inhibition [31] and heightened extrasynaptic NMDA receptor signaling, which is closely associated with neuronal death [17,32,33,34].

To explore the interplay between sex, synaptic changes, and NMDA receptor mislocalization, we employed an integrative approach combining correlative and invasive techniques. Using the APP/PS1 double transgenic mouse model of AD and age- and sex-matched wild-type (WT) controls, our study aimed to identify potential sex-specific differences in synaptotoxicity and investigate the role of extrasynaptic GluN2B-containing NMDA receptors in AD pathogenesis. These findings could provide critical insights for developing targeted therapeutic strategies.

## 2. Results

### 2.1. Sex-Specific Impact on Spatial Memory Impairments in APP/PS1 Mice

The memory performance of male and female APP/PS1 mice was evaluated using two distinct but complementary paradigms, each tailored to target spatial memory processes particularly vulnerable to AD pathology [35]. First, we used the spatial discrimination paradigm in the eight-arm radial maze, which required learning and remembering the spatial location of three constantly baited arms (Figure 1a). Acquisition curves demonstrated that the performance of male and female APP/PS1 and WT mice improved over the 10-day training period, as indicated by a progressive decline in the daily mean number of total memory errors (Figure 1b). However, APP/PS1 mice exhibited slower acquisition in the spatial discrimination task compared to WT mice (Figure 1b). Importantly, the observed spatial discrimination impairment was similar across male and female APP/PS1 mice, indicating no sex-dependent differences (Figure 1c).

To exclude the potential influence of prolonged learning periods on the observed lack of difference between male and female APP/PS1 mice, we further investigated the impact of sex on spatial recognition memory using the Y-maze (Figure 1d). WT mice demonstrated a significant preference for the novel arm during the retrieval phase. In contrast, APP/PS1 mice failed to discriminate the new arm (Figure 1e). This impairment was not sex-specific, as both male and female APP/PS1 mice exhibited deficits compared to their respective WT (Figure 1f). Furthermore, this transgene-associated impairment could not be attributed to a deficit in APP/PS1 in processing visual cues or an alteration in their innate preference for novelty, since these mice were able to recognize the novel arm when the ITI was reduced to 1 min, as shown in female mice (Appendix A). These findings suggest that both male and female APP/PS1 mice exhibit impairments in forming longer-lasting spatial memories.

Twenty-four hours after Y-maze testing, all mice were euthanized to assess amyloid-beta (Aβ) accumulation. We focused our analysis on the hippocampus, a brain region critically involved in spatial memory formation and affected early in AD [36,37]. Our results revealed a significant difference in the Aβ_1–42_ levels between APP/PS1 and WT mice (Figure 1g), and we found no significant effect of sex on the transgenic effect (Figure 1g). Thus, a robust Aβ_1–42_ accumulation in APP/PS1 mice occurred regardless of sex.

### 2.2. Sex-Specific Impact on Synaptic Marker Alterations in APP/PS1 Mice

To investigate whether the memory impairments exhibited by APP/PS1 mice are linked to alterations in synaptic markers, we analyzed the total hippocampal protein expression, focusing on markers associated with learning and memory processes. First, we examined the presynaptic marker synaptophysin, often indicative of synaptic loss, a common feature in AD [38,39,40]. However, no significant differences were found in the synaptophysin expression between male APP/PS1 and WT mice (Supplementary Appendix A). Given the critical roles of postsynaptic density protein 95 (PSD-95) and NMDA receptors in memory processing [41,42,43,44], we further evaluated the expression of PSD-95 and NMDA receptor subunits, including GluN1, GluN2B, and GluN2A. No significant changes were observed between male APP/PS1 and WT mice in GluN1 or GluN2B expression (Supplementary Appendix A). A significant reduction in PSD-95 protein expression was observed in APP/PS1 mice compared to WT controls when both males and females were considered together (Figure 2a,b), along with a decrease in GluN2A expression (Figure 2a,d). However, the genotype x sex interaction failed to reach significance for these two brain markers, suggesting that the observed synaptic changes were not sex-dependent (Figure 2c,e). Finally, we also assessed calcium/calmodulin-dependent protein kinase II (CaMKII), which is highly concentrated at synaptic sites and, through its interaction with GluN2B, plays a critical role in learning and memory [45,46,47]. We found no alteration associated with AD pathology (Supplementary Appendix A).

As a scaffolding protein, PSD-95 facilitates the proper anchoring of NMDA receptors at postsynaptic sites, ensuring their effective positioning for synaptic signaling and plasticity [48]. Consequently, changes in PSD-95 levels can disrupt NMDA receptors’ trafficking and/or their synaptic localization without affecting their overall expression. To explore this further, and given that GluN2B subunits are localized in both synaptic (postsynaptic density, PSD) and extrasynaptic (non-PSD) compartments, we aimed to determine whether the subcellular localization of these subunits might be altered despite there being no changes in the overall receptor levels. To examine potential shifts in the NMDA receptor distribution between these compartments, we utilized a fractionation protocol [49]. First, to confirm the validity of our fractionation approach, we qualitatively assessed the expression of PSD-95 and synaptophysin. This quality control step was essential to validate our samples for the further examination of the subcellular location of NMDA receptors. As shown in Figure 3a, PSD-95 was highly enriched in the PSD fractions of WT mice but undetectable in the non-PSD fraction. In contrast, the presynaptic marker synaptophysin was primarily localized to the non-PSD fraction. Since the PSD fraction is a primary target for Aβ_1–42_ [50,51], we quantified the Aβ_1–42_ levels in these fractions in females. The concentration of soluble Aβ_1–42_ was markedly elevated in the PSD fraction of APP/PS1 mice (Appendix A). These results indicate that soluble Aβ_1–42_ preferentially accumulates within the PSD of APP/PS1 mice, suggesting that this accumulation may interfere with synaptic receptor trafficking. Next, we examined the expression of the GluN2A and GluN2B subunits. Both subunits were detected in the PSD fraction, with only GluN2B present in the non-PSD fraction (Figure 3b).

We then quantified the proportion of GluN2B in both compartments in WT and APP/PS1 mice, with both sexes pooled together. As shown in Figure 3c, APP/PS1 mice exhibited a significant reduction in synaptic GluN2B and an increase in extrasynaptic GluN2B compared to age-matched WT controls. This shift in extrasynaptic GluN2B was independent of sex (Figure 3d,e). Our observations suggest that alterations in the scaffolding protein PSD-95 are able to influence the trafficking of NMDA receptors, potentially contributing to an increase in the extrasynaptic NMDA receptor pool in APP/PS1 mice. However, we cannot rule out the possibility that other mechanisms might also be responsible for this effect. This implies that the activation of extrasynaptic NMDA receptors could potentially mediate the impairment in learning and memory processes in the hippocampus of APP/PS1 mice.

### 2.3. DL-TBOA Alters Spatial Discrimination Performance in Adult Mice

To further validate the above-mentioned hypothesis, we took advantage of the functional properties of the glutamate uptake inhibitor DL-TBOA, which induces glutamate spillover, subsequently leading to an enhanced recruitment of extrasynaptic NMDA receptors [30,52,53]. Given the potential for DL-TBOA to induce seizures, we first carried out pilot experiments to determine the optimal DL-TBOA concentration that could be injected into the hippocampus without generating side effects. Higher concentrations (2.5 and 1.5 nmol) led to seizures within one hour post infusion (Appendix A). In contrast, injections of 0.75 nmol and 0.5 nmol of DL-TBOA did not induce seizures. However, the 0.75 nmol dose caused irreversible impairments in memory formation with mice being unable to re-learn the location of the newly accessible arm in the Y-maze even seven days after the DL-TBOA injection (Appendix A). This inability to learn was not due to repeated training in the Y-maze, as aCSF-injected mice were able to form short-term memories, performing well during repeated testing (Appendix A). Thus, the 0.75 nmol dose of DL-TBOA likely induced irreversible neuronal alterations, a phenomenon that was not observed with the lower injected dose of 0.5 nmol (Appendix A).

We therefore bilaterally injected DL-TBOA (0.5 nmol; 0.5 µL) into the hippocampus of male adult C57BL/6J mice 40 min before the encoding phase in the Y-maze (Figure 4a,b). As we observed no sex effect in the proportion of extrasynaptic NMDA receptors between male and female mice, the impact of extrasynaptic NMDA receptors on memory processes was evaluated only in male mice. As expected, a strong preference for the unvisited arm was observed during the test phase in aCSF-injected control mice (Figure 4c,d). In contrast, DL-TBOA-injected mice failed to discriminate the novel arm (Figure 4c,d). To rule out irreversible excitotoxic lesions, DL-TBOA-injected mice were re-tested seven days later. As shown in Figure 4d, these mice were able to discriminate the new arm, similar to aCSF-injected controls. Thus, the observed deleterious effect of DL-TBOA on memory performance appears to be primarily related to a time-limited alteration in synaptic transmission rather than nonspecific irreversible neuronal death secondary to excitotoxicity.

To evaluate the functional contribution of GluN2B subunits to the DL-TBOA-induced memory impairment, we next performed bilateral injections of DL-TBOA and ifenprodil (12 nmol), a selective GluN2B antagonist, into the hippocampus of C57BL/6J mice 40 min before the encoding phase of the Y-maze paradigm (Figure 4a). Only mice co-injected with ifenprodil were able to discriminate the novel arm, indicating that ifenprodil effectively blocked the effect of glutamate spillover triggered by DL-TBOA (Figure 4d). Importantly, this TBOA-induced impairment was memory-specific, as there were no confounding effects on the total exploration of the open arm during the encoding phase assessed by the distance traveled (Figure 4e) and the number of open-arm entries (Figure 4f). Collectively, our data suggest that the synaptic accumulation of Aβ may impair the trafficking of GluN2B-containing NMDA receptors, leading to the presence of a higher number of GluN2B subunits in the extrasynaptic compartment. This imbalance in the localization of GluN2B subunits—favoring the extrasynaptic over the synaptic compartment—may contribute, at least in part, to the altered memory profile observed in the APP/PS1 mice.

Altogether, our findings suggest that the synaptic accumulation of Aβ potentially impairs the trafficking of GluN2B-containing NMDA receptors within the synaptic compartment, leading to an increased presence of GluN2B in the extrasynaptic compartment. Importantly, this effect is independent of sex, as it has been observed similarly in both female and male APP/PS1 mice.

## 3. Discussion

Our study offers new insights into the interplay between NMDA receptor organization, biological sex, and AD-induced synaptotoxicity through a proteomic analysis of the whole hippocampus and distinct neuronal compartments in APP/PS1 double knock-in mice. We observed a comparable amyloid burden and cognitive impairments in both male and female mice and roughly similar synaptic alterations. The downregulation of PSD-95 and the synaptic accumulation of Aβ_1–42_ were closely linked to a marked reduction in GluN2A subunits and the mislocalization of GluN2B subunits to the extrasynaptic compartment. Furthermore, using intrahippocampal injections of DL-TBOA, we showed that the predominant activation of extrasynaptic GluN2B-containing NMDARs impaired spatial recognition memory in the Y-maze by transiently disrupting synaptic transmission. This highlights the critical role of these receptors in memory dysfunction. Our findings suggest a potential mechanism where Aβ destabilizes synaptic organization by reducing PSD-95 levels, leading to a decrease in synaptic GluN2A subunits and an increase in extrasynaptic GluN2B subunits. This reorganization likely contributes to the memory deficits characteristic of AD, underscoring the importance of synaptic receptor trafficking in disease progression.

Numerous studies in humans have demonstrated a sex-specific effect on the prevalence of AD, with women disproportionately affected compared to men [2,3]. This disparity has been attributed to various factors, including hormonal differences, such as the decline in estrogen after menopause, as well as potential genetic and lifestyle influences that increase the risk in women. While certain murine models of AD suggest a worsening of pathology in females [54,55], we did not observe any profound differences in the progression of pathological markers or cognitive impairments between males and females in APP/PS1 mice. This could be explained by the fact that, in APP/PS1 mice, the pathology develops well before the changes in estrogen levels, a key factor in accelerating the disease [56]. The menopause-associated decline in estrogen, a hormone with neuroprotective properties, is known to reduce synaptic protection and may accelerate disease progression in women [57].

Interestingly, the expression levels of NMDA receptors differ significantly between male and female rodents, particularly within specific brain regions. These sex-specific variations have important implications for neurological function and pharmacological responsiveness. For instance, female rats exhibit higher expression levels of GluN1 and GluN2B NMDA receptor subunits in the hippocampus compared to males [58,59], which may influence memory formation and spatial processing. Additionally, in female rats, NMDA receptor density within the hippocampus varies across the estrous cycle. During estrus, female display significantly lower NMDA receptor density in the oriens and radiatum layers of the CA1, CA2, and CA3 subregions compared to males, a pattern also observed during diestrus [58]. These sex-dependent differences in NMDA receptor expression are associated with altered sensitivity to NMDA receptor blockade in females [44,60] and increased susceptibility to glutamate-induced neurotoxicity [61]. Such receptor-level variations likely contribute to differences in cognitive flexibility and learning between male and female rodents [62], potentially providing a mechanistic basis for the sex-specific effects observed in AD.

The loss of synapses and reductions in synaptic markers are well documented in both the early and late stages of AD and are strongly correlated with cognitive deficits [5,63,64]. While presynaptic changes have been consistently observed, the expression levels of postsynaptic markers such as PSD-95 have shown inconsistent results, with some studies reporting reductions and others reporting increases [26,29,65,66,67,68]. In our study, we confirmed PSD-95 downregulation in APP/PS1 mice of both sexes, supporting the role of PSD-95 in learning impairment [25,69]. This suggests that PSD-95 could serve as a molecular marker for behavioral deficits and potentially as a therapeutic target, to mitigate synaptic damage in AD, irrespective of sex [25]. PSD-95 is more than a structural protein; it plays a critical role in regulating NMDA receptor trafficking [70]. The early loss of PSD-95 in AD likely disrupts synaptic receptor function initially and, over time, impacts synaptic structure. Consistent with this, we observed alterations in the expression and subcellular distribution of NMDA receptor subunits. Similar to findings in human patients with mild cognitive impairment [65], our AD mouse model exhibited the downregulation of both PSD-95 and synaptic GluN2A subunits. Interestingly, while the total GluN2B expression remained unchanged, its redistribution shifted toward the extrasynaptic compartment. This redistribution of GluN2B could be linked to PSD-95 downregulation, given PSD-95’s role in supporting NMDA receptors’ trafficking to synapses from the endoplasmic reticulum [71,72,73] and maintaining them at the synapse by preventing internalization [74]. The differential effects of PSD-95 downregulation on the GluN2A and GluN2B subunits might be explained by variations in their intracellular trafficking kinetics, which are driven by distinct C-terminal domain motifs. These motifs allow GluN2B-containing NMDARs greater mobility, lateral diffusion, faster internalization, and recycling, while GluN2A-containing receptors are internalized more slowly and are more likely to fuse with degradative endosomes [74,75,76]. As GluN2A-containing NMDA receptors are less recyclable, they may be more sensitive to PSD-95 reductions than GluN2B subunits [77].

A key consequence of GluN2A downregulation is a shift in the GluN2A/GluN2B ratio, which is essential for memory formation and the maintenance of synaptic stability [42]. This ratio plays a critical role in several neuronal processes, including spine motility, synaptogenesis [78], and the regulation of synaptic plasticity thresholds [79]. An abnormal shift in the GluN2A/GluN2B ratio has functional implications, as GluN2B-containing NMDA receptors exhibit longer decay time constants [80] and carry more calcium, which can lead to oxidative stress and neuronal death [81], commonly observed in pathological conditions [82]. We confirmed a decrease in synaptic GluN2B subunits associated with AD, and we also demonstrated an increase in their extrasynaptic expression, consistent with previous findings [17,34,83,84,85]. This suggests a redistribution of GluN2B-containing NMDA receptors, as the total GluN2B expression remained unchanged. Notably, this mislocalization has been recently validated in AD patients’ tissue using similar approaches, further underscoring its translational significance [86]. Some studies, such as those by Snyder et al. [17], did not detect a difference in extrasynaptic NMDA receptor staining after Aβ exposure, suggesting that Aβ-induced NMDA receptor endocytosis may specifically target synaptic NMDA receptors rather than extrasynaptic ones. These findings are supported by work from Li and colleagues [34]. This discrepancy between these results and ours could reflect differences in the disease stage being modeled, with exogenous Aβ application representing early-stage processes, while our 9-month-old APP/PS1 mice reflect a later stage of AD pathology. A detailed time-course study of GluN2B alterations in APP/PS1 mice could help resolve this issue. Nevertheless, based on our experiments involving the overactivation of extrasynaptic NMDA receptors, we propose that the mislocalization of GluN2B-containing NMDA receptors, as observed in APP/PS1 mice, is a significant contributor to glutamatergic dysfunction which drives memory impairments in AD. The fact that we observed this mislocalization in both male and female subjects strengthens the case for targeting this mechanism as a promising therapeutic avenue for both sexes.

To date, most studies have investigated the role of extrasynaptic receptors in synaptic plasticity using long-term potentiation (LTP) and long-term depression (LTD) models in hippocampal slices. Similar to the effects of exogenous Aβ application [87], the functional recruitment of extrasynaptic NMDA receptors has been shown to impair LTP development and induce or enhance LTD [52,88]. In our study, we used cerebral injections of DL-TBOA to mimic the imbalance between synaptic and extrasynaptic NMDA receptor responses. We demonstrated that the activation of extrasynaptic receptors impaired cognitive performance in mice via a GluN2B-dependent mechanism. These findings suggest that the mislocalization of GluN2B receptors and/or an imbalance between synaptic and extrasynaptic signaling may disrupt memory encoding in AD. This hypothesis is supported by studies showing that GluN2B antagonists can counteract the effects of Aβ [89]. Furthermore, our data provide a mechanistic explanation for the therapeutic benefit of memantine, one of the first approved drugs for treating mid- to late-stage AD, which selectively blocks extrasynaptic NMDA receptors [33,90]. The increase in extrasynaptic signaling has been attributed to both NMDA receptor redistribution, as shown in our study, and glutamate spillover caused by glial transporter dysfunction, as suggested by others [91].

Although our results did not pinpoint the precise mechanisms by which Aβ disrupts PSD-95 and/or NMDA receptor localization, we observed a significant synaptic accumulation of Aβ in the PSD fraction but not in the non-PSD fraction in females. This finding concurs with previous in vitro studies [18,92] and mirrors observations in human AD patients. Notably, the amount of synaptic dimeric Aβ oligomers in synaptosomes has been inversely correlated with Mini-Mental State Examination scores [93], and more recently, both monomeric and dimeric Aβ species from soluble and detergent extracts have been associated with dementia in AD cases [94]. Several neuronal receptors have been proposed as Aβ binding partners, including α7-nicotinic receptors [17], glutamatergic receptors [95,96,97], insulin receptors [98], and cellular prion proteins [99,100]. Many of these receptors are localized within the PSD and may account for the preferential targeting of Aβ. Moreover, these receptors often reside in cholesterol-rich microdomains within the plasma membrane, known as lipid rafts, which are crucial for Aβ production and aggregation [101]. The aberrant accumulation of Aβ in these lipid rafts may directly alter the mobility of key proteins, such as glutamate receptors, as reported for the mGluR5 subtype [95]. Thus, the accumulation of Aβ oligomers at excitatory synapses may serve as an early initiating factor in AD pathology.

This study has several limitations. While our findings suggest comparable alterations in both males and females, a more systematic analysis of all the markers studied in both sexes, with a similar proportion of male and female animals, would be necessary to strengthen our conclusions and ensure comparable statistical power between the sexes. Additionally, validating these findings in another transgenic model, or ideally in postmortem tissues from AD patients, would provide valuable insight into whether the observed effect is model-specific or generalizable to human pathology. Another aspect of our study that warrants further investigation is the influence of sex on the cellular localization of Aβ and its various species [102,103] and aggregation forms [104]. While Aβ accumulation was primarily investigated in females across different cellular compartments in this work, future research incorporating both sexes would offer valuable insights into potential sex differences.

Understanding how alterations in NMDA receptor signaling contribute to AD pathology has been a key focus of research for decades. However, therapeutic strategies aimed at correcting NMDA receptor dysfunction have largely failed, potentially because of a predominant focus on the modulation of NMDA receptor channel properties. Given the central role of NMDA receptor ionotropic signaling throughout the body, such approaches often lead to significant adverse effects. This highlights the need for more nuanced strategies targeting the synaptic trafficking and compartmentalization of NMDA receptors rather than just their ion channel functions. We found that the overall trajectory of synaptic dysfunction remained consistent between sexes, highlighting the potential for sex-independent therapeutic approaches in future research and interventions to reduce synaptotoxicity in AD.

## 4. Materials and Methods

### 4.1. Animals

The double transgenic APPswe/PS1dE9 (APP/PS1) mice were generated by crossing male Tg2576 expressing human APP with Swedish mutation mice with female PS1dE9 transgenic mice, which express a human presenilin 1 (PS1) variant lacking exon 9 and associated with familial AD. The APPswe line was derived from a C57BL/6xSJL background (Taconic Inc., Germantown, NY, USA), while the PS1dE9 line was derived from a C57BL/6J background (Jackson Laboratory, Bar Harbor, ME, USA). To further minimize any potential confounding effects related to the strain background and housing differences, we used sex-matched wild-type (WT) littermates for comparison in our experiments. Both transgenes were under the control of the mouse prion protein (PrP) promoter, facilitating expression primarily in neurons. All mice were heterozygous for the transgenes, and genotypes were confirmed via polymerase chain reaction on tail biopsies.

The mice were aged 9 to 10 months old at the start of the experiment. Both male and female APPswe/PS1dE9 mice and their age- and sex-matched wild-type (WT) littermates were used. Additionally, adult (3–4 months) male C57BL/6J mice (Janvier Lab, Laval, France) were included to assess the effects of intrahippocampal injections of DL-TBOA, DL-TBOA + ifenprodil, or artificial cerebrospinal fluid (aCSF). Two weeks prior to behavioral experiments, the mice were housed individually. The experimental procedures complied with official European Guidelines for the care and use of laboratory animals (directive 2010/63/UE) and were approved by the ethical committee of the University of Bordeaux (protocol A50120159) and followed the ARRIVE guidelines for animal research.

### 4.2. Guide Cannula Implantations into the Hippocampus

Adult male C57BL/6J mice were bilaterally implanted with guide cannulas under deep general anesthesia. Each mouse was anesthetized using Vetflurane (induction at 4%; maintenance at 1.5–2%) and secured in a Kopf stereotaxic frame. Prior to surgery, the mice were treated with buprenorphine and an anti-inflammatory drug to ensure analgesia during and after the procedure. To prevent dry eyes, a lubricating gel was applied as needed during the procedure. Body temperature was closely monitored and maintained at a stable level of 37 °C.

Following anesthesia, the scalp was incised and retracted to expose the skull. Craniotomies were performed directly above the target regions, and the dura was carefully cut to reveal the cortex. Holes were drilled in the skull before lowering the stainless-steel guide cannulas to the following coordinates: posterior −2 mm, mediolateral −/+1.4 mm, and dorsoventral −1 mm relative to the bregma point. After placement, the guide cannulas were secured with dental cement. A topical analgesic was applied immediately after surgery and as needed thereafter.

### 4.3. Intracerebral Injections into Freely Moving Mice

All infusions were conducted in a preparation room separate from the behavioral testing area. During the infusion process, animals were gently restrained by the experimenter. Infusion cannulas, extending 1 mm beyond the tips of the guide cannulas, were inserted into the guides. Bilateral infusions were performed simultaneously using two Hamilton syringes connected to the infusion cannulas via propylene tubing. The syringes were operated by a Harvard Apparatus precision syringe pump, which delivered 0.5 µL of DL-TBOA, DL-TBOA + ifenprodil, or aCSF to each hippocampus over a period of 2 min. The infusion cannulas were then left in place for an additional 90 s to allow for drug diffusion before removal. Prior to behavioral testing, mice were familiarized with all aspects of the infusion procedure, except that the injection cannulas were not connected to the infusion pump. This step was taken to minimize stress levels that could interfere with subsequent memory assessments.

### 4.4. Behavioral Testing

#### 4.4.1. Y-Maze

The Y-maze apparatus was constructed from gray Plexiglas and featured three arms positioned at 120° angles to one another. Each arm measured 8 cm × 30 cm × 15 cm (width × length × height). The arms were randomly designated as follows: the “start arm”, where the mouse was initially placed; the “open arm”, accessible during both the encoding and retrieval phases; and the “novel arm”, which was blocked during encoding but open during retrieval (Figure 1e). The testing procedure comprised two trials, separated by an inter-trial interval (ITI), to evaluate spatial recognition memory. The first trial (encoding) lasted 5 min, allowing the mouse to explore only the start and open arms, with the novel arm being blocked. Following a 10 or 1 min ITI [105], the recognition trial began, during which all three arms were accessible. The exploration pattern of each mouse was analyzed by comparing the time spent in each arm, assessing novelty versus familiarity. Video recordings were analyzed to determine the time spent in each arm, the number of entries, and the total distance traveled. The successful discrimination of the novel arm from the two familiar arms during the first 2 min of exploration served as an index of spatial recognition memory [106]. Memory performance was expressed as the percentage of time spent in the novel arm, calculated as follows: ((seconds in novel arm)/(seconds in previously visited arms + seconds in the novel arm) × 100). Mice were tracked using the Noldus video-tracking system.

#### 4.4.2. Eight-Arm Radial Maze

Mice underwent spatial discrimination testing in an 8-arm radial maze. Each arm measured 62 cm in length and 12 cm in width, radiating from a central platform 32 cm in diameter. Two weeks prior to testing, mice were housed individually with ad libitum access to food and water. Daily handling was conducted, and free-feeding weights were recorded over three consecutive days. Before starting the discrimination task, mice were gradually food-restricted over 8 days until they reached 85–90% of their free-feeding weight. This restriction was maintained throughout the testing period by adjusting their daily food intake. During testing trials, arm choices were video-tracked using Poly software version 4.4.1 (Imetronic, Marcheprime, France). Sucrose pellets (one 20 mg pellet per arm; Dustless Precision Pellets, BioServ, San Diego, CA, USA) were used as rewards and placed at the ends of the selected arms. The training phase spanned 12 days and consisted of 2 days of habituation followed by 10 days of testing. During the habituation phase, mice freely explored the maze, with all arms accessible and baited. Each daily trial concluded when the animal collected all 8 rewards. On the following day, food-restricted mice began spatial discrimination testing. Each mouse was assigned a fixed set of 3 baited arms out of 8, with arm separations of 45°, 90°, and 135° (e.g., arms 1-2-4, 2-3-5, etc.; Figure 1a). Mice completed 6 daily trials with an ITI of 1 min over 10 consecutive days. At the start of each trial, the mouse was placed on the central platform with all doors closed. After one minute, all doors opened simultaneously, allowing the mouse to freely choose among the 8 arms of the maze. Once the mouse returned to the platform after visiting an arm, the doors closed for 4 s before reopening, which prevented clockwise or counterclockwise sweeping strategies (consecutive arm visits, e.g., 1-2-3-4…). A trial ended when the mouse collected the pellets from all three baited arms, after which it remained on the central platform until the next trial. Total memory errors included repeated visits to baited arms as well as visits to non-baited arms.

### 4.5. Euthanasia and Brain Extraction

For biochemical analysis, mice were euthanized using a lethal intraperitoneal injection of pentobarbital and subsequently decapitated. The brains were extracted, and the hippocampi were placed in ice-cold PBS solution and stored at −80 °C until further analysis. Each hippocampus was frozen separately for independent treatment, either for fractionation or total protein extraction. To control cannula placements after behavioral testing, mice were deeply anesthetized using a lethal intraperitoneal injection of pentobarbital, and their whole brains were collected and frozen at −80 °C. Coronal sections (30 μm thick) were generated using a cryostat, focusing on areas proximal to the guide and cannula tracts, and mounted on PLUS slides (Thermo Fisher Scientific, Villebon-sur-Yvette, France) until dry. Cannula placements were then confirmed under a light microscope. Since the injection sites were accurately located within the dorsal hippocampus for all mice, none were excluded from the data analyses.

### 4.6. Biochemical Analysis

Hippocampi from APP/PS1 and WT littermates were homogenized using an 18 G needle in a buffer containing 20 mM HEPES, 0.15 mM NaCl, 1% Triton X-100, 1% deoxycholic acid, and 1% SDS (Buffer IV, pH 7.5). The samples were incubated for 1 h at 4 °C, followed by centrifugation at 10,000× *g* for 15 min. The supernatant was collected as the total protein extract. Subcellular fractionation was performed as follows: tissues were resuspended in a cold buffer containing 0.32 M sucrose and 10 mM HEPES, pH 7.4 (Buffer I). Homogenates were cleared by centrifugation at 1000× *g* for 10 min, and the resulting supernatants were further centrifuged at 12,000× *g* for 20 min to concentrate the crude membrane fraction. This fraction was then washed in a buffer containing 4 mM HEPES and 1 mM EDTA, pH 7.4 (Buffer II), and centrifuged at 12,000× *g* for 20 min. The pellet was resuspended in 20 mM HEPES, 100 mM NaCl, and 0.5% Triton X-100 (Buffer III, pH 7.2) and incubated for 15 min before being centrifuged at 12,000× *g* for 20 min. The resulting supernatant represented the non-postsynaptic density (non-PSD) membrane fraction. The pellet was further solubilized in 20 mM HEPES, 0.15 mM NaCl, 1% Triton X-100, 1% deoxycholic acid, and 1% SDS (Buffer IV, pH 7.5) for 1 h and then centrifuged at 10,000× *g* for 15 min to isolate the postsynaptic density (PSD) fraction. All buffers were supplemented with protease and phosphatase inhibitors (Sigma, L’Isle d’Abeau, France), and protein concentration was measured using a BCA assay kit (Pierce, Thermo Fisher Scientific, Villebon-sur-Yvette, France).

### 4.7. Western Blotting

The protein concentrations of brain homogenates were determined using the Bradford assay and normalized to 5–20 µg per sample. Proteins were separated by SDS-PAGE on Tris–glycine precast 4–15% gels (Bio-Rad, Marnes-La-Coquette, France) and transferred to polyvinylidene difluoride (PVDF) membranes using the Transblot Turbo^®^ system and Transblot Turbo RTA^®^ transfer kit (Bio-Rad, Marnes-La-Coquette, France). Membranes were blocked in TTBS (0.05% Tween 20, 200 mM NaCl, 10 mM Tris, pH 7.4) with 5% non-fat dry milk for 1 h at room temperature, followed by incubation with primary antibodies diluted in 5% non-fat dry milk–TTBS at 4 °C overnight with gentle agitation. Secondary fluorescent-conjugated antibody incubation was performed for 1 h at room temperature. After three washes with TTBS and one with PBS, the membranes were scanned using a Licor Aerius automated infrared imaging system (LI-COR, Bad Homburg, Germany) according to the manufacturer’s instructions. Quantification was based on band intensity using the software (Odyssey Fc) provided with the imaging system. Due to limited sample availability, membranes were stripped and reused up to two times. For antibody removal, membranes were incubated in stripping buffer (0.2 M glycine, 0.1% SDS, 1% Tween 20, pH 2.2) for 5–10 min. Membranes were then washed with TTBS, scanned to confirm antibody removal, and reblocked with TTBS containing 5% non-fat dry milk before incubation with a new primary antibody. Actin and GluN1 proteins were used as a loading control for female and male samples of total proteins, respectively. Actin was used as a loading control for PSD and non-PSD fractions’ protein quantification. The primary antibodies used included the following: anti-synaptophysin (Millipore, Molsheim, France, 1 μg/mL), anti-PSD95 (Millipore, 2 μg/mL), anti-GluN2A (Millipore, 0.1 μg/mL), anti-GluN2B (Millipore, 1 μg/mL), anti-Actin (Sigma-Aldrich, St. Louis, MO, USA, 0.8 μg/mL), anti-CaMKII (Santa Cruz Biotechnology, Dallas, TX, USA, 1 μg/mL), and anti-NR1 (Santa Cruz, 1 μg/mL). The secondary antibodies were Goat anti-rabbit (IR Dye^®^ 800CW, LI-COR, Bad Homburg, Germany) or Goat anti-mouse (IR Dye^®^ 680RD, LI-COR).

### 4.8. Quantification of Aβ_1–42_ Peptide by ELISA

The concentration of human Aβ_1–42_ peptide was measured by ELISA (Amyloid-Beta 42 Human ELISA Kit, Thermo Fisher Scientific, Villebon-sur-Yvette, France), according to the manufacturer’s instructions, in equal amounts of total, PSD, or non-PSD hippocampal extracts from APP/PS1 mice and WT littermates. This kit specifically detects soluble forms of human Aβ_1–42_ peptides with negligible cross-reactivity with Aβ_1–40_ forms. Aβ_1–42_ concentrations were determined by comparing sample absorbance at 450 nm to a standard curve (0–250 pg/mL) using a microplate reader. The experimenters were blinded to the animal groups during the Western blot and ELISA analysis.

### 4.9. Statistical Analyses

Statistical analyses were conducted using GraphPad Prism version 10.0 (http://www.graphpad.com; access on 1 January 2024). Data were tested for normal distribution using Kolmogorov–Smirnov tests. For normally distributed data, unpaired or paired *t*-tests and one-way, two-way, or three-way ANOVAs were used to identify significant differences between groups. To identify outliers, the Rout test was used. No animals were excluded from the analysis. Values of *p* < 0.05 were considered as statistically significant. The full list of statistical analyses is shown in Supplementary Appendix A.

## Figures and Tables

**Figure 1 ijms-26-01737-f001:**
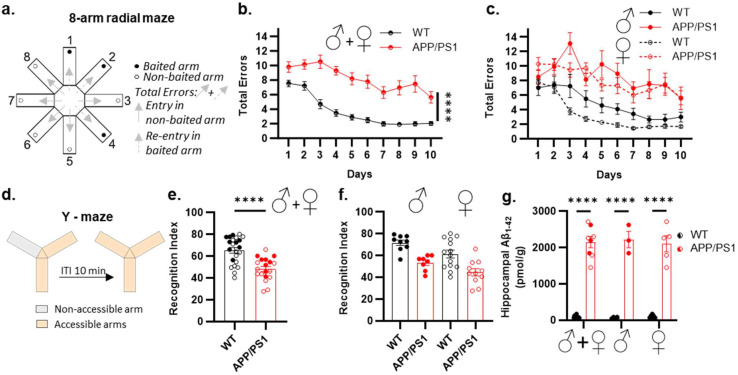
Memory impairment and amyloid-β accumulation in female and male APP/PS1 mice. (**a**) Experimental design of 8-arm radial maze testing. Mice needed to locate and remember the spatial location of the three constantly baited arms of the maze. Total errors were scored as entry into non-baited arms plus re-entry into baited arms. (**b**) Learning curves of male and female APP/PS1 and WT. (**c**) Learning curves showing male and female APP/PS1 and WT separately. (**d**) Experimental design: Y-maze task, with 10 min inter-trial interval (ITI). (**e**,**f**) Performance of the APP/PS1 mice was severely impaired compared to wild-type (WT) littermates (**e**), and no sex difference was observed (**f**). (**g**) Amyloid-β_1–42_ (Aβ_1–42_) accumulation in the hippocampus is significant at the age of 9–10 months old in both female and male APP/PS1 mice. Data points are individual mice with mean ± SEM. Statistical significance was calculated using two-way ANOVA (**b**,**f**,**g**), three-way ANOVA (**c**), or the *t*-test (**e**), **** *p* < 0.0001.

**Figure 2 ijms-26-01737-f002:**
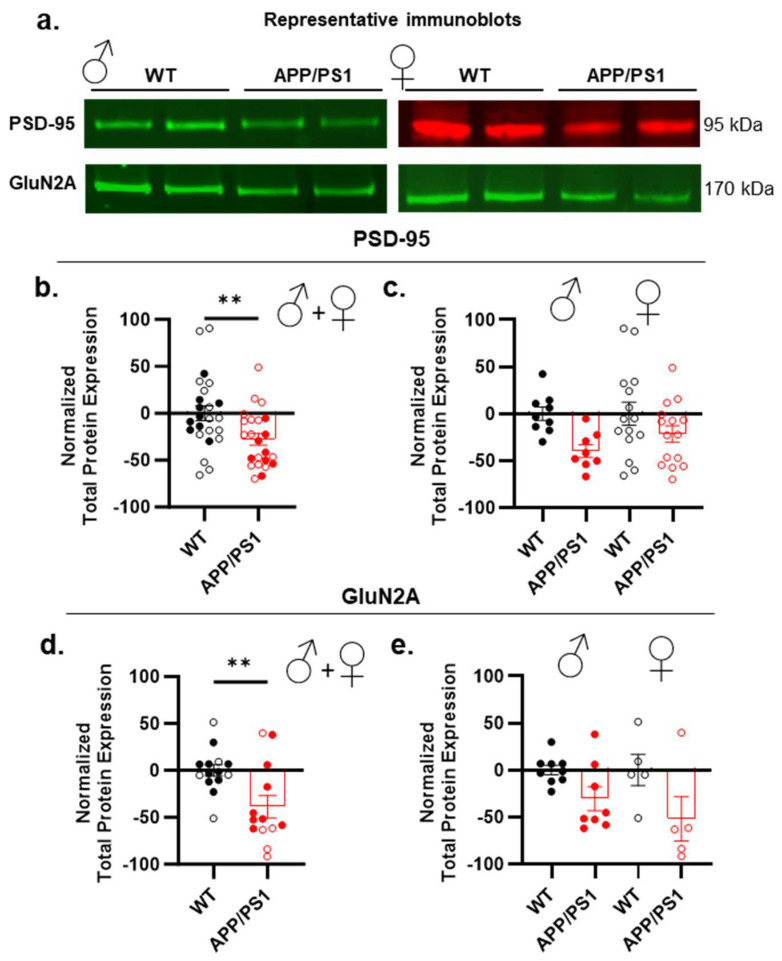
Hippocampal expression of cerebral markers in male and female APP/PS1. (**a**) Representative Western blots of the different brain markers analyzed in APP/PS1 and WT mice. (**b**,**c**) Changes in PSD-95 protein observed between WT and APP/PS1 mice (**b**) are not sex-dependent (**c**). (**d**,**e**) Changes in GluN2A subunits observed between genotypes (**d**) show no difference between female and male APP/PS1 and WT (**e**). Data points are individual mice with mean ± SEM. Statistical significance was calculated using the *t*-test (**b**,**d**) and two-way ANOVA (**c**,**e**), ** *p* < 0.01.

**Figure 3 ijms-26-01737-f003:**
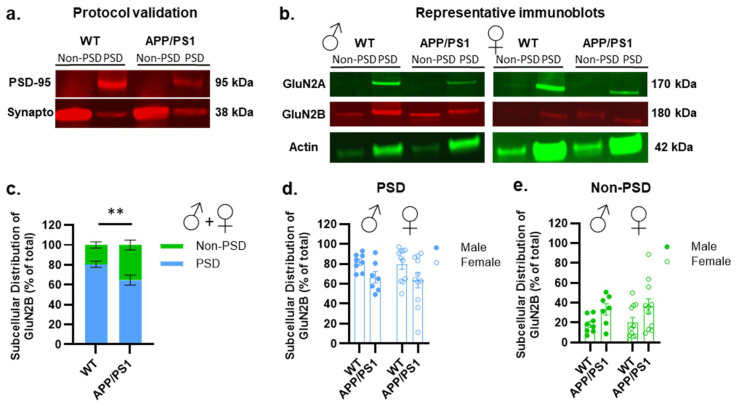
Postsynaptic (PSD) and non-postsynaptic (non-PSD) hippocampal expression of cerebral markers in male and female APP/PS1. (**a**) Representative Western blots of the different brain markers analyzed in APP/PS1 and WT mice used for the validation of the fractionation protocol. (**b**) Representative Western blots of the different brain markers analyzed in male and female APP/PS1 and WT mice. (**c**–**e**) Changes in the level of GluN2B subunits of NMDA receptors are observed between PSD and non-PSD compartments between APP/PS1 mice compared to the WT (**c**), with no sex effect observed for both PSD (**d**) and non-PSD (**e**) compartments. Data points are individual mice with mean ± SEM. Statistical significance was calculated using two-way ANOVA (**c**–**e**), ** *p* < 0.01.

**Figure 4 ijms-26-01737-f004:**
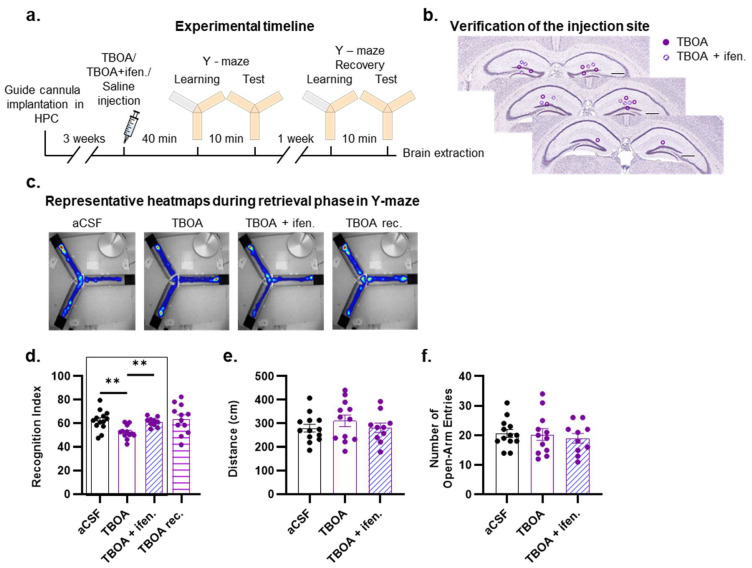
DL-TBOA-induced glutamate spillover alters spatial discrimination performance in mice. (**a**) Experimental timeline. (**b**) Coronal diagrams of mouse brain sections showing injection sites in dorsal hippocampus. Scale bar: 0.5 mm. (**c**) Representative heatmaps during retrieval phase in Y-maze for each group of mice. (**d**) Spatial memory was impaired in mice injected with DL-TBOA compared to aCSF, an effect that was time-dependent, as treated mice performed at the level of aCSF-injected mice after 7 days (TBOA rec.). The deleterious effect of DL-TBOA was blocked by the simultaneous injection of GluN2B subunit blocker ifenprodil (TBOA + Ifen.). (**e**,**f**) Injection of DL-TBOA and DL-TBOA + ifenprodil did not affect mouse exploratory activity, such as distance passed (**e**) and number of open-arm entries (**f**) during the initial phase of training. Data points are individual mice with mean ± SEM. Statistical significance was calculated using one-way ANOVA (**d**–**f**), ** *p* < 0.01.

## Data Availability

All data are available in this manuscript or on request from the corresponding author.

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
