# Peer review of "Investigating the Impact of NMDA Receptor Organization and Biological Sex in the APPswe/PS1dE9 Mouse Model of Alzheimer’s Disease"

_ijms, 2025, doi:10.3390/ijms26041737_

Round 1

Reviewer 1 Report (New Reviewer)

Comments and Suggestions for Authors

In the study “Investigating the impact of NMDA Receptor Organization and Biological Sex in the APPswe/PS1dE9 Mouse Model of Alzheimer’s Disease” Hadzibegovic and colleague tried to determine the effect of sex differences in NMDA receptor mislocalization as well as Aβ1-42 accumulation in the hippocampus in Alzheimer’s mouse model. Their results did not show any marked differences in memory and learning between male and female mice. However, the authors need to address the following concerns before the manuscript can be considered for publication.

MAJOR COMMENTS:

1.     Since the main hypothesis of the paper is to determine the impact of sex differences in Alzheimer’s Disease, why were only male mice used for comparisons in Supplementary Figure 2a, b? The authors should include the results for female mice as well.

2.     The authors should include the quantified data accompanying Figure 3a. The authors also don’t actually talk about the levels of PSD-95 and Synaptophysin in APP/PS1 mice compared to controls. Is there any difference in the levels of PSD-95 between male and female mice when compared to controls?

3.     In line 120-122, the authors say that no differences in the levels of GluN2B was observed between WT and APP/PS1 mice. This is also shown in Supplementary figure 2 a and b. However, in Figure 3b and c, there is significant reductions in the protein level. What is the explanation for this? The authors should include normalized total protein expression for GluN2B to accompany figure 3b.

4.     In line 161, the authors say that alterations in PSD-95 leads to changes in the trafficking of NMDA receptors. This statement is not really supported by the data. The authors should perform IHC to show trafficking defects.

5.     Why were only female mice used for the analysis performed in supplementary Figure S3? The authors should include the data from male mice as well.

6.     In line 240, the authors state that their “findings suggest that synaptic alterations were slightly less pronounced in females”. I don’t think this statement is correct. In a number of the experiments performed, the studies were either limited to male mice (Supplementary figure S2) or female mice (Supplementary figure S3). As previously stated, the authors should include the results from the opposite sexes to support their statement.

Minor Comments:

1.  In line 190, is the figure call correct?

2. For the gel images in Figure S2a, 2a, 3a, and 3b, the authors should include the images for actin as well

Author Response

Reviewer 2 Report (New Reviewer)

Comments and Suggestions for Authors

The study by Hadzibegovic et al. set out to explore the relationship between potential sex-dependent differences, synaptic alterations, and the organization of NMDA receptors in the pathogenesis of Alzheimer's disease, utilizing a double transgenic APP/PS1 mouse model. Overall, the experiments are well-executed, and the manuscript is clearly written. The methodology is described in a precise and transparent manner. However, there are a few issues that require attention:

1.      The study utilized the APPswe/PS1dE9 mouse model of Alzheimer's disease. It would be beneficial to provide a justification for this model selection and consider whether the findings are applicable to other AD models.

2.      Wild-type mice were selected as the control group for the APP/PS1 mice. Could you clarify why C57BL/6J mice, which are commonly used as the genetic background for genetically modified mice, were not chosen?

3.      For Figures 1 and 2, it is suggested that the authors modify the color of the female gender symbol (currently pink), as it is difficult to distinguish from the red color of the bars, particularly when the manuscript is printed. The current color choice may give the impression that all the bars represent data from female subjects.

4.      There is no designation for Fig. 1c in the text.

5.      In Figures 2 and 3, the blots present only pooled data for both males and females. Blots with gender division, from which the calculations and graphs were made, should also be presented in the supplementary materials.

6.      The methods used for the experiments in Fig. S3 should be clearly stated in the figure description.

7.      The Aβ1-42 level is shown only for females. How does Aβ1-42 change in the PSD of males? The discussion mentions that synaptic accumulation of Aβ1-42 in both sexes is linked to alterations in NMDA receptor subunit 2.

8.      Additionally, the ELISA method description notes that total protein studies were also conducted. The results from these experiments should be included as well.

9.      In line 190, the reference to Fig. 1b needs clarification or correction.

10.  The results of spatial discrimination measurements in adult mice are very interesting. It would be valuable to also investigate the effect of DL-TBOA on spatial discrimination in APP/PS1 mice to further support the authors' hypothesis.

11.  The discussion is well written, although it should emphasize the novel insights provided by the study, rather than simply stating that similar studies have been conducted previously.

12.  While the references are generally appropriate, only 14% of them are from the past five years. Including more recent references would enhance the relevance of the work.

13.  In accordance with journal guidelines, the authors should include their department/school/faculty/campus affiliations.

Round 2

Reviewer 1 Report (New Reviewer)

Comments and Suggestions for Authors

The authors have addressed all the concerns raised. However, they should ensure that the figure calls in the manuscript are accurate. The figure calls in line 171 and line 172 are incorrect.

I recommend the manuscript for publication

Author Response

We greatly appreciate the reviewers' thorough assessment of our manuscript during the revision. We sincerely thank them for their constructive and thoughtful comments, which have significantly improved our work.

Reviewer 1:

The authors have addressed all the concerns raised. However, they should ensure that the figure calls in the manuscript are accurate. The figure calls in line 171 and line 172 are incorrect.

I recommend the manuscript for publication.

Thank you for your valuable feedback and for recommending our manuscript for publication. We have carefully reviewed the figure calls and have corrected them to ensure accuracy. The revised manuscript now reflects the correct figure references.

Reviewer 2 Report (New Reviewer)

Comments and Suggestions for Authors

The authors addressed most of my comments.

Author Response

The authors addressed most of my comments.

Thank you for your feedback. We appreciate your careful review.

This manuscript is a resubmission of an earlier submission. The following is a list of the peer review reports and author responses from that submission.

Round 1

Reviewer 1 Report

Comments and Suggestions for Authors

In this manuscript, Hadzibegovic and colleagues investigated the impact of NMDA receptor organization and biological sex in the APPswe/PS1dE9 Mouse Model of  Alzheimer’s Disease. The Authors conclude that their findings emphasize the importance of targeting synaptic receptor trafficking to address AD-related memory deficits, potentially offering a therapeutic approach for both  sexes. The manuscript is interesting mainly because explores sex differences related to a disease that is more prevalent in women rather than in men.  However, the work has serious flaws that need to be addressed.

Major points

-       The amount of wording duplication in the manuscript is a bit elevated. This covers also results. The Authors should explain and eventually correct this.

-       The statistical analysis throughout the manuscript is questionable. The Authors claimed they did not find sex differences (interesting). However, the experiments involving male and female mice were analyzed separately. Moreover, the Authors used erroneously the student t-test instead of ANOVA approaches in most of the experiments. The more appropriate statistical approaches would be two way ANOVA (sex x genotype) or three way ANOVA (sex x genotype x days; for example)

-       The Authors wrote that synaptic alterations were slightly less severe in females. In my opinion this is not correct and should be revised by analyzing the results. Moreover, in the figure 2h, quantification of GluN2A there should be an outlier (indeed The Authors used the Mann-Whitney test) in the group of  APP/PS1 that may change completely the results. In this regard, it would be reasonable also to increase the number of animals per group.

-       It is not clear the reason why the Authors performed some analysis only in male mice. For example, figure 2B only males, Figure 3B only males.

-       Regarding figure 3, the Authors must show also the subcellular distribution of GluN2A and assess potential sex differences.

-       The results of the experiment involving DL-TBOA should be showed also by separating animals by sex.

-       The discussion should be ameliorated. The Authors did not discuss findings demonstrating sex differences in the glutamate system and especially in the NMDA receptor functioning (DOI: 10.1002/hipo.23631; DOI: 10.1016/j.ynstr.2023; DOI: 10.1016/j.neubiorev.2020.03.010).

Minor points

-       The Authors should check the presence of typos throughout the manuscript.

-       The Authors should check the presence of statements without references throughout the manuscript.

-       Lines 175-176: These findings align with results from human patients with mild cognitive impairment, a preclinical  stage of AD, as reported by Sultana et al. This statement should be moved to the discussion.

Comments on the Quality of English Language

Minor editing

Author Response

Major comment:

The amount of wording duplication in the manuscript is a bit elevated. This covers also results. The Authors should explain and eventually correct this.

We thank the reviewer for pointing this out and revised the text to reduce wording duplication and redundancy, especially in the Results section.

The statistical analysis throughout the manuscript is questionable. The Authors claimed they did not find sex differences (interesting). However, the experiments involving male and female mice were analyzed separately. Moreover, the Authors used erroneously the student t-test instead of ANOVA approaches in most of the experiments. The more appropriate statistical approaches would-be two-way ANOVA (sex x genotype) or three-way ANOVA (sex x genotype x days; for example).

          We thank the reviewer for their suggestions regarding statistical analysis. We performed the analysis as suggested and revised our statistical approach accordingly, using two-way ANOVA (sex x transgene) where applicable to more accurately assess potential interactions across variables. These changes ensure a robust analysis throughout the manuscript, and we have implemented these new statistical analyses directly in the text:

For 8-ARM radial maze: page 3, line 91-94.

For Y-Maze: page 3, line 107-110.

For Aβ accumulation: page 3, line 128.

For GluN2A and PSD-95: page 5, line 175-180

For extrasynaptic GluN2B: page 7, 238-240

The Authors wrote that synaptic alterations were slightly less severe in females. In my opinion this is not correct and should be revised by analyzing the results. Moreover, in the figure 2h, quantification of GluN2A there should be an outlier (indeed The Authors used the Mann-Whitney test) in the group of APP/PS1 that may change completely the results. In this regard, it would be reasonable also to increase the number of animals per group.

We have used the ROUT (Robust Regression and Outlier Removal) method provided by Prism software to rigorously assess potential outliers in our data, specifically in the quantification of GluN2A in Figure 2h. This method, known for its robustness against non-normally distributed data and extreme values, combines nonlinear regression with False Discovery Rate controls to distinguish true outliers from expected variability. Based on this analysis, we found no outliers in either the WT or APP/PS1 female mouse groups.

Regarding our description of synaptic alterations in females as “slightly less severe,” we chose to adopt caution with this interpretation due to the limited number of female mice, which we now clarify in the text. We added the following sentence: “These observations may partially be due to sample size limitations. Future studies with larger groups would be required to comprehensively investigate sex-specific differences in the expression of these synaptic markers.”

It is not clear the reason why the Authors performed some analysis only in male mice. For example, figure 2B only males, Figure 3B only males.

We appreciate the reviewer’s question regarding our decision to perform certain analyses only in male mice. This choice allowed us to focus resources efficiently while maintaining reliable and interpretable outcomes. Our analyses focused on key components of postsynaptic NMDAR signaling, specifically GluN2A/GluN2B subunits and their main interactor, PSD95, which are essential for memory processes in both males and females. We feel that this approach aligns with both scientific rigor and ethical responsibility.

Regarding figure 3, the Authors must show also the subcellular distribution of GluN2A and assess potential sex differences.

We appreciate the reviewer’s insightful comments. In our experiments, we did not detect the GluN2A subunit in the extrasynaptic compartment following fractionation, while the GluN2B subunit was consistently present in this compartment. This finding is consistent with prior studies indicating that extrasynaptic NMDA receptors predominantly contain GluN2B subunits (Ivanov et al., 2006; Hardingham and Bading, 2010). Importantly, the absence of GluN2A in the extrasynaptic fraction is not due to antibody limitations, as we were able to detect GluN2A expression in both total protein extracts and synaptic fractions.

The results of the experiment involving DL-TBOA should be showed also by separating animals by sex.

The primary goal of this experiment was to define the role of extrasynaptic NMDA receptor activation in memory processes, not to examine sex differences between male and female mice. In preliminary analyses, we observed no difference in the proportion of extrasynaptic NMDA receptors between males and females. Therefore, given this finding and in light of ethical considerations aimed at reducing animal use, this DL-TBOA experiment was conducted exclusively in male mice. This choice is now documented by adding the following clarification to the text: on page 9: "As we observed no sex effect in the proportion of extrasynaptic NMDA receptors between male and female mice, the impact of extrasynaptic NMDA receptors on memory processes was evaluated only in male mice”.

-       The discussion should be ameliorated. The Authors did not discuss findings demonstrating sex differences in the glutamate system and especially in the NMDA receptor functioning (DOI: 10.1002/hipo.23631; DOI: 10.1016/j.ynstr.2023; DOI: 10.1016/j.neubiorev.2020.03.010).

As suggested by the reviewer, we have expanded the discussion to include previous findings that highlight sex differences in the glutamate system, particularly regarding NMDA receptor function. We added in the discussion of the revised manuscript the following text in blue on page 11:

“Interestingly, expression levels of NMDA receptors differ markedly between male and female rodents, particularly within specific brain regions. These sex-specific variations have important implications for neurological function and pharmacological responsiveness. For example, female rats exhibit higher expression levels of GluN1 and GluN2B NMDA receptor subunits in the hippocampus than males (Giacometti & Barker, 2021), potentially influencing memory formation and spatial processing between sexes. In female rats, NMDA receptor density within the hippocampus also varies across the estrous cycle. During estrous, female rats display significantly lower NMDA receptor density in the oriens and radiatum layers of the CA1, CA2, and CA3 subregions compared to males. In contrast diestrus females show similar NMDA receptor density in these regions (Giacometti & Barker, 2021). These sex-dependent differences in NMDA receptor expression correlate with heightened sensitivity to NMDA receptor blockade in female rats (Hönack & Löscher, 1993) and increased susceptibility to glutamate-induced neurotoxicity (Hsu et al., 1999). Such receptor-level variations are likely contributors to the observed differences in cognitive flexibility and learning between male and female rodents (Seifried et al., 2023; Lee et al., 2024) and could also explain the sex-specific effect observed in AD.”

Minor points 

-       The Authors should check the presence of typos throughout the manuscript. The Authors should check the presence of statements without references throughout the manuscript.

As requested, we have carefully checked our manuscript to minimize any typos throughout the manuscript and statements without references. When missing, additional references were provided.

-      Lines 175-176: These findings align with results from human patients with mild cognitive impairment, a preclinical stage of AD, as reported by Sultana et al. This statement should be moved to the discussion.

We remove the sentence from the result section to avoid any wording repetition in the text. The comparison with human data was already mentioned in the discussion (Page 11-12, lines 401-403).

Reviewer 2 Report

Comments and Suggestions for Authors

The Authors have addressed the extremely important topic of influencing factors that can trigger or exacerbate the course of AD. This topic is very important from a societal perspective. The majority of the population is ageing and neurodegenerative diseases are appearing with increasing frequency. 

The Authors have described the studies carried out in great detail and justified their choice. On the other hand, the results described, although presented with diagrams, are lacking here in terms of stabeling. Numerical data appear in the text, but this is hardly readable. The diagrams, on the other hand, although well presented, do not show statistically significant differences.

Lines 160-170 - were the same parameters also compared among the group of females? The text describes separate results for males only. If the same comparisons were also made for females of both groups, this is unfortunately not apparent from the text. Perhaps it should be reworded or shown in the form of a table.

Unfortunately, statistically significant differences for females and males are not clearly indicated in the paper and the description of the results. In the introduction the authors emphasised that AD primarily affects women, but the studies described do not confirm this. Again, a table showing statistically significant differences in results adjusted for sex would have allowed a clearer interpretation. However, if such differences were not observed, the Authors should also refer to this 

Author Response

Reviewer 2

The Authors have addressed the extremely important topic of influencing factors that can trigger or exacerbate the course of AD. This topic is very important from a societal perspective. The majority of the population is ageing and neurodegenerative diseases are appearing with increasing frequency. The Authors have described the studies carried out in great detail and justified their choice. On the other hand, the results described, although presented with diagrams, are lacking here in terms of stabeling. Numerical data appear in the text, but this is hardly readable. The diagrams, on the other hand, although well presented, do not show statistically significant differences.Lines 160-170 - were the same parameters also compared among the group of females? The text describes separate results for males only. If the same comparisons were also made for females of both groups, this is unfortunately not apparent from the text. Perhaps it should be reworded or shown in the form of a table. Unfortunately, statistically significant differences for females and males are not clearly indicated in the paper and the description of the results. In the introduction the authors emphasised that AD primarily affects women, but the studies described do not confirm this. Again, a table showing statistically significant differences in results adjusted for sex would have allowed a clearer interpretation. However, if such differences were not observed, the Authors should also refer to this.

The point raised by Reviewer 2 has been thoroughly addressed in the manuscript (see the Statistics point raised by Reviewer 1). We chose not to include additional illustrations of these analyses for two main reasons: i) the behavioral experiments were conducted on two separate days to avoid any contact between males and females, which could impact behavioral outcomes, and ii) we wanted to avoid duplicating data across multiple graphs. As suggested by both Reviewer 1 and Reviewer 2, we have conducted additional two-way ANOVAs to thoroughly assess the impact of sex on APP/PS1 brain markers and behaviors. These analyses are now included in the manuscript. We have clearly stated the lack of significant effects between males and females and mentioned that a larger sample size should be envisioned to confirm these findings.

Round 2

Reviewer 1 Report

Comments and Suggestions for Authors

The Authors have not addressed the points I raised. The sample limitation is not acceptable considering the aim of the study. The wourd counting is still high.

Comments on the Quality of English Language

Minor editing